# On Non-Linear operators for Geometric Deep Learning

**Grégoire Sergeant-Perthuis**
Univ. Artois, UR 2462, Laboratoire de
Mathématiques de Lens (LML)
F-62300 Lens, France
& OURAGAN team, Inria Paris & IMJ-PRG
Paris, France.
gregoireserper@gmail.com

**Jakob Maier**
INRIA, DI/ENS, PSL
Paris

**Joan Bruna**
Courant Institute of Mathematical Sciences
New York University
New York

**Edouard Oyallon**
MLIA - Machine Learning
and Information Access
Sorbonne Université, CNRS,
ISIR, F-75005 Paris, France

## Abstract

This work studies operators mapping vector and scalar fields defined over a manifold $\mathcal{M}$, and which commute with its group of diffeomorphisms $\text{Diff}(\mathcal{M})$. We prove that in the case of scalar fields $L^p_\omega(\mathcal{M}, \mathbb{R})$, those operators correspond to point-wise non-linearities, recovering and extending known results on $\mathbb{R}^d$. In the context of Neural Networks defined over $\mathcal{M}$, it indicates that point-wise non-linear operators are the only universal family that commutes with any group of symmetries, and justifies their systematic use in combination with dedicated linear operators commuting with specific symmetries. In the case of vector fields $L^p_\omega(\mathcal{M}, T\mathcal{M})$, we show that those operators are solely the scalar multiplication. It indicates that $\text{Diff}(\mathcal{M})$ is too rich and that there is no universal class of non-linear operators to motivate the design of Neural Networks over the symmetries of $\mathcal{M}$.

## 1 Introduction

Given a physical domain $\mathcal{M}$ and measurements $f : \mathcal{M} \to \mathcal{Y}$ observed over it, one is often interested in processing *intrinsic* information from $f$, i.e. consistent with the *symmetries* of the domain. Let $M$ denote an operator, it can be seen as a non-linear operator acting on measurements. In words, if two measurements $f$, $\tilde{f} = g.f$ are related by a symmetry $g$ of the domain, like a rigid motion on an observed molecular compound, we would like our processed data $M(f)$ and $M(\tilde{f})$ to be related by the same symmetry — thus that $M(g.f) = g.M(f)$ or equivalently that $M$ commutes with the symmetry transformation of the domain. The study of operators that satisfy such symmetry constraints has played a long and central role in the history of physics and mathematics, motivated by the inherent symmetries of physical laws. More recently, such importance has also extended to the design of machine learning systems, where symmetries improve the sample complexity [25, 3]. For instance, Convolutional Neural Networks build translation symmetry, whereas Graph Neural Networks build permutation symmetry, amongst other examples coined under the 'Geometric Deep Learning' umbrella [5, 4].

36th Conference on Neural Information Processing Systems (NeurIPS 2022).

Lie groups of transformations are of particular interest, because there exists a precise and systematic framework to build such intrinsic operators. Indeed, for a locally compact group $G$, it is possible to define a Haar measure which is invariant to the action of $G$ [2]; then a simple filtering along the orbit of $G$ allows to define a class of *linear* operators that commute with the group action. Examples of locally compact groups are given by specific Lie groups acting on $\mathbb{R}^d$, such as the translations or the rotations $O_d(\mathbb{R})$. Often these Lie groups $G$ only act on a manifold $\mathcal{M}$, and one tries to average along the orbit induced by $G$. Note that it is possible, beyond invariance, to linearize more complex groups of variability like diffeomorphisms $\text{Diff}(\mathcal{M})$ [7].

While the description of such linear intrinsic structures is of central mathematical importance and forms the basis of Representation theory [30], in itself is not sufficient to bear fruit in the context of Representation *learning* using Neural Networks [12]. Indeed, linear operators do not have the capacity to extract rich information needed to solve challenging high-dimensional learning problems. It is therefore necessary to extend the systematic construction and classification of intrinsic operators to the non-linear case.

With that purpose in mind, our work aims at studying the class of (*non-linear*) operators $M$ which commute with the action of the group $\text{Diff}(\mathcal{M})$, the diffeomorphisms over $\mathcal{M}$. This approach will lead to a natural class of non-linear intrinsic operators. Indeed, any group $G$ of symmetries is, by definition, a subgroup of $\text{Diff}(\mathcal{M})$, and thus commutes with such $M$ [24]. Consequently, obtaining a non-linear invariant to a symmetry group $G$ could be done by using a cascade of interlacing non-linear operators which commute with $\text{Diff}(\mathcal{M})$ and linear operators which commute with $G$.
A notable example of linear operators that are covariant to the Lie group of translations is a given by the convolutions along the orbit of the group. These can be constructed thanks to the canonical Haar measure [32]. However, such an approach fails for infinite dimensional groups, like our object of interest: contrary to Lie groups, $\text{Diff}(\mathcal{M})$ is not locally compact and it is thus not possible to define a Haar measure on this group.

Our first contribution is to demonstrate that the *non-linear* operators which act on vector fields (elements of $L^p_\omega(\mathcal{M}, T\mathcal{M})$) and which commute with the group of diffeomorphisms, are actually just scalar multiplications. This implies that $\text{Diff}(\mathcal{M})$ is too rich to obtain non-trivial operators. Our second contribution is to demonstrate that *non-linear* operators acting on signals in $L^p_\omega(\mathcal{M}, \mathbb{R})$ are pointwise non-linearities. This fills a gap in the results of [7], and *a fortiori* justifies the use of point-wise non-linearities in geometric Deep Learning [4].

Let us remark that the study of equivariant operators that take as input vector fields is motivated by the use of Neural Networks in physics, in particular for dynamical systems such as fluid dynamics [8]. For example, one subject of interest in hydrodynamics is how a vector field of velocities evolves; the time evolution of such field is described by a partial differential equation (PDE), the Navier-Stokes equations, in which Neural Networks found recent applications and it is more generally the case of other PDE [31].

Our paper is structured as follows: Sec. 2 introduces the necessary formalism, that we use through this paper: in particular, we formally define the action of diffeomorphism. Then, we state and discuss our theorems in Sec. 3.1 and sketch their proofs in Sec. 3.2. Rigorous proofs of each statement can be found in the Appendix.

## 2 Problem Setup

### 2.1 Related work and motivation

In this section, we discuss the notion of intrinsic operators, invariant and covariant non-linear operators and linear representation over standard symetry groups. Then, we formally state our objective.

**Intrinsic Operators**   As discussed above, in this work we are interested in *intrinsic* operators $M : L^p(\mathcal{M}, E) \to L^p(\mathcal{M}, E)$, where $\mathcal{M}$ is a Riemannian manifold, and $E = \mathbb{R}$ or $E = T\mathcal{M}$, capturing respectively the setting of scalar signals and vector fields over $\mathcal{M}$. $L^p(\mathcal{M}, \mathbb{R})$ is the space of scalar function $f : \mathcal{M} \to \mathbb{R}$ which $p$-th power is integrable, similarly $L^p(\mathcal{M}, T\mathcal{M})$ is the space of sections of the tangent bundle of $\mathcal{M}$ (denoted $T\mathcal{M}$), $f : \mathcal{M} \to T\mathcal{M}$, which norm $\|f\| : \mathcal{M} \to \mathbb{R}$ is in $L^p(\mathcal{M}, \mathbb{R})$. Here the notion of 'intrinsic' means that $M$ is consistent with an equivalence class induced by a symmetry group $G$ in $L^p(\mathcal{M}, E)$: if $f, \tilde{f} \in L^p(\mathcal{M}, E)$ are related by a transformation

$g \in G$ (in which case we write $f = g.\tilde{f}$), then $M(f) = g.M(\tilde{f})$. Naturally, a stronger equivalence class imposes a stronger requirement towards $M$, and consequently restrains the complexity of $M$. We now describe the plausible techniques used to design such operators $M$.

**GM-Convolutions**   The notion of $GM$-convolutions [34] is an example of linear covariant operators which commute with the reparametrization of a manifold. In practice, this implies that the weights of a $GM$-convolution are shared and the action of $GM$-convolutions is local – two properties that facilitate implementation and point out the similarity with Lie groups. Another example of symmetry group corresponds to the isometry group of a Riemaniann manifold, whose pushforward preserves the tensor metric. In this case, it is well known that isometries [33] are the only diffeomomorphism which commute with a manifold Laplacian. Thus, any *linear* operators which commute with isometries is stabilized by Laplacian's eigenspaces. However, little is known on the *non-linear* counterpart of the symmetry-covariant operators. In this work, we characterize *non-linear* operators which commute with $\text{Diff}(\mathcal{M})$. We will see that such operators are intrinsically defined by $\text{Diff}(\mathcal{M})$ and could be combined with any linear operators covariant with a symmetry group $G$.

**Non-linear operators**   It has been shown that Convolutional Neural Networks are dense in the set of *non-linear* covariant operators [35]. The recipe of the corresponding proof is an extension of the proof of the universal approximation theorem [14]. The Scattering Transform [6, 23] is also an example of a well-understood non-linear operator which corresponds to a cascade of complex wavelet transforms followed by a point-wise modulus non-linearity. This representation provably linearizes small deformations.

**Compact Lie Groups**   In the context of geometric Machine Learning [5], there are several relevant notions of equivalence. For instance, we can consider a compact Lie Group $G$ acting on $\mathcal{M}$, and an associated representation in $\mathcal{F} = \{f : \mathcal{M} \to \mathbb{R}\}$: Given $g \in G$ and $f \in \mathcal{F}$, then $g.f(x) \triangleq f(g^{-1}.x)$ for $x \in \mathcal{M}$. We then consider $f \sim \tilde{f}$, related by this group action: $\tilde{f} = g.f$ for some $g \in G$. The operators $M$ which are compatible with such group action are referred as being $G$-equivariant (or covariant to the action of $G$) in the ML literature [13, 4]. Such groups are typically of finite and small dimension, e.g. the Euclidean transformations of $\mathcal{M} = \mathbb{R}^d$, with $d = 2$ for computer vision applications, or $d = 3$ for computational biology/chemistry applications. In this case, it is possible to characterize all *linear* intrinsic operators $M$ as group convolutions [20], leading to a rich family of non-linear intrinsic operators by composing such group convolutions with element-wise non-linear operators, as implemented in modern Neural Networks. We highlight that stability to symetries via non-linear operators finds useful application, in particular for flat manifolds [7].

**Isometries**   Riemanian manifolds $\mathcal{M}$ come with a default equivalence class, which is given by isometries. $T_u\mathcal{M}$ denotes the tangent vector space of $\mathcal{M}$ at point $u \in \mathcal{M}$. If $m_u : T_u\mathcal{M} \times T_u\mathcal{M} \to \mathbb{R}$ denotes the Riemannian metric tensor at point $u \in \mathcal{M}$, a diffeomorphism $\psi : \mathcal{M} \to \mathcal{M}$ is an isometry if $g_u(v, w) = g_{\psi(u)}(d\psi_u(v), d\psi_u(w))$ for any $u \in \mathcal{M}$ and $v, w \in T_u\mathcal{M}$. In words, isometries are changes of variables that preserve the local distances in the domain. The ensemble of all isometries forms a Lie Group which is locally compact [27]. In this case, one can also build a rich class of intrinsic operators by following the previously explained 'blueprint', namely composing linear intrinsic operators with element-wise non-linearities. As a representative example, the Laplace-Beltrami operator of $\mathcal{M}$ only depends on intrinsic metric properties [33]: as said above, isometries preserve the invariant subspaces of a Laplacian.

**Beyond Isometries**   While isometries are the 'natural' transformations of the geometric domain, they cannot express high-dimensional sources of variability; indeed, if $\mathcal{M}$ is a $d$-dimensional complete connected Riemannian manifold, its isometry group has dimension at most $d(d + 1)/2$ [10]. This raises the question whether one can characterize intrinsic operators relative to a broader class of transformations. Another class of important symmetries corresponds to the ones which are gauge invariant, i.e. which leads to transformations which preserve the change of parametrization and which are used in [11, 34] through the notion of $G$-structure.

In this work, we consider the class of transformations given by $\text{Diff}(\mathcal{M})$, the diffeomorphisms over $\mathcal{M}$. As shown in the Appendix, compactly supported deformations $\psi : \mathcal{M} \to \mathcal{M}$ define bounded linear operators $L_\psi$ acting on $L^p(\mathcal{M}, E) \to L^p(\mathcal{M}, E)$, and constitute a far broader class

of transformations than isometries. Our proof is mainly based on the use of compactly supported diffeomorphisms.

Our objective is to characterize the (non-linear) operators $M$ such that

$$\forall \phi \in \mathrm{Diff}(\mathcal{M}), L_\phi M = M L_\phi \, .$$

In other words, we aim to understand continuous operators $M$ that commute with deformations. We will show that such operators are act locally and that they can be descriped explicitly, with simple formula. The commutation condition is visualized in the following diagram:

$$
\begin{array}{ccc}
f & \xrightarrow{\ L_\phi\ } & g \\
\downarrow{\scriptstyle M} & \circlearrowleft & \downarrow{\scriptstyle M} \\
Mf & \xrightarrow{\ L_\phi\ } & Mg
\end{array}
$$

## 2.2 Notations

We will now formally introduce the mathematical objects of interest in this document. Let $(\mathcal{M}, g)$ be an orientable, connected, Riemannian manifold, of finite dimension $d \in \mathbb{N}^*$. Let $T\mathcal{M}$ denote the tangent bundle of $\mathcal{M}$, i.e. the union of tangent spaces at points $u \in \mathcal{M}$. $T^*\mathcal{M}$ is the cotangent bundle of $\mathcal{M}$. $g \in \Gamma(T^*M \otimes T^*M)$ is a section of symmetric definite positive bilinear forms on the tangent bundle of $M$. It is common to denote $\Gamma B$ the collection of sections of a bundle $B$; $\bigwedge^n T^*M$ for $n \leq d$ is the bundle of $n$-linear alternated forms of $\mathcal{M}$, and $\Gamma(\bigwedge^n T^*M)$ is the space of section of this vector bundle over $\mathcal{M}$.

For $A \subseteq \mathcal{M}$, we denote $\overline{A}$ its closure; $1_A$ is the indicator function of $A$, i.e. which takes value 1 if $x \in A$ and 0 otherwise. $\mathcal{B}(u, r)$ denotes the ball of radius $r$ around $u \in \mathcal{M}$. Any two vectors $v, v_1 \in V$ in a pre-Hilbert space (with a scalar product $\langle , \rangle$) are orthogonal, denoted $v \perp v_1$, when $\langle v, v_1 \rangle = 0$.

Fix $p \in [1, +\infty[$. Any volume form $\omega \in \Gamma(\bigwedge^d T^*M)$ defines a (positive) measure on the orientable Riemannian manifold $\mathcal{M}$; the total volume of $\mathcal{M}$ is $\omega(\mathcal{M}) := \int_{\mathcal{M}} 1 d\omega$. Let us define $L_\omega^p(\mathcal{M}, T\mathcal{M})$, the space of $L^p$ vector fields, defined as the subspace of measurable functions $f : \mathcal{M} \to T\mathcal{M}$ such that $f(u) \in T_u M$ almost everywhere and

$$\|f\|_p^p \triangleq \int_{u \in \mathcal{M}} g_u(f(u), f(u))^{\frac{p}{2}} \, d\omega(x) < +\infty \, . \tag{1}$$

We will also consider $L_\omega^p(\mathcal{M}, \mathbb{R})$ the space of measurable scalar functions (fields) $f : \mathcal{M} \to \mathbb{R}$ that fulfill

$$\|f\|_p^p \triangleq \int_{u \in \mathcal{M}} |f(u)|^p \, d\omega(u) < +\infty \, . \tag{2}$$

We may write $\| \cdot \|$ instead of $\| \cdot \|_p$ when there is no ambiguity. For a $C^\infty$ diffeomorphism $\phi \in \mathrm{Diff}(\mathcal{M})$, we will consider the action of $L_\phi : L_\omega^p(\mathcal{M}, T\mathcal{M}) \to L_\omega^p(\mathcal{M}, T\mathcal{M})$ which we define for for any $f \in L_\omega^p(\mathcal{M}, \mathbb{R})$ as

$$L_\phi f(u) \triangleq d\phi(u)^{-1}.f(\phi(u)) \, .$$

Note that this action is contravariant:

$$L_{\psi \circ \phi} f(u) = d(\psi \circ \phi)^{-1}.f(\psi \circ \phi(u)) = L_\phi L_\psi f(u)$$

For scalar function $f \in L_\omega^p(\mathcal{M}, \mathbb{R})$, we define the action of $\phi$ via

$$L_\phi f(u) \triangleq f(\phi(u)) \, .$$

Let $A$ be a measurable set of $\mathcal{M}$ and $f \in L^p(\mathcal{M}, E)$, $f 1_A$ is the product of $f$ with $1_A$, i.e. $f 1_A$ is equal to $f$ on $A$ and $0$ elsewhere. In what follows we introduce 'constant' fields over an open set, they are denoted $c 1_U$ with $U$ an open subset of $\mathcal{M}$. For scalar fields, a 'constant' scalar field $f(u)$

is equal to the same constant $c \in \mathbb{R}$ for any $u \in U$. On the other hand, 'constant' vector fields $f1_U$ are vector fields over $U$ for which there is a chart from $U$ to an open subset of $\mathbb{R}^d$, in which for any $u \in U$ $f(u)$ is equal to a constant vector $c \in \mathbb{R}^d$; in the vector case we say that the vector field $f1_U$ can be straightened.

This latter operator is also contravariant. If there is no ambiguity, we will use the same notation $L_\phi$, whether we apply it to $L_\omega^p(\mathcal{M}, \mathbb{R})$ or $L_\omega^p(\mathcal{M}, T\mathcal{M})$. We might sometimes refer to $L_\omega^p(\mathcal{M}, \mathbb{R})$ or $L_\omega^p(\mathcal{M}, T\mathcal{M})$ as $L^p(\mathcal{M}, \mathbb{R})$ or $L^p(\mathcal{M}, T\mathcal{M})$. Throughout the article we restrict ourselves to $\phi$ such that $L_\phi$ is a bounded operator. Write $\text{supp}(\phi) = \{u, \phi(u) \neq u\}$ for the support of $\phi$ and say that $\phi$ has a compact support if $\text{supp}(\phi)$ is compact. We denote by $\text{Diff}_c(\mathcal{M}) \subset \text{Diff}(\mathcal{M})$ the set of compactly supported diffeomorphisms. Recall that since $\mathcal{M}$ is second-countable, $\mathcal{C}_c^\infty(\mathcal{M})$ is dense in $L_\omega^p(\mathcal{M}, \mathbb{R})$ and $\mathcal{C}_c^\infty(\mathcal{M}, T\mathcal{M})$ is dense in $L_\omega^p(\mathcal{M}, T\mathcal{M})$. Finally, denote by $O_d(\mathbb{R})$ the set of unitary operators on $\mathbb{R}^d$. Throughout the article, we might not write explicitly that equalities hold almost everywhere, since this is the default in $L^p$ spaces.

As mentioned earlier, compactly supported diffeomorphisms lead to continuous operators, which is made rigorous by the following lemma whose proof is in the appendix.

**Lemma 1.** *If $supp(\phi)$ is compact, then $L_\phi$ is bounded.*

## 3 Main theorems

In this section we present our main results. We first show that any (non-linear) deformation-equivariant operator acting on scalar fields must be point-wise (Theorem 1), and then establish that any deformation-equivariant operator acting on vector fields corresponds to a multiplication by a scalar (Theorem 2).

### 3.1 Theorem statements

Now, we are ready to state our two main theorems:

**Theorem 1** (Scalar case). *Let $\mathcal{M}$ be a connected and orientable manifold of dimension $d \geq 1$. We consider a Lipschitz continuous operator $M : L_\omega^p(\mathcal{M}, \mathbb{R}) \to L_\omega^p(\mathcal{M}, \mathbb{R})$, where $1 \leq p < \infty$. Then,*

$$\forall \phi \in Diff(\mathcal{M}) : \ ML_\phi = L_\phi M$$

*is equivalent to the existence of a Lipschitz continuous function $\rho : \mathbb{R} \to \mathbb{R}$ that fulfills*

$$M[f](m) = \rho(f(m)) \quad a.e.$$

*In that case, we have $\rho(0) = 0$ if $\omega(\mathcal{M}) = \infty$.*

**Theorem 2** (Vector case). *Let $\mathcal{M}$ be a connected and orientable manifold of dimension $d \geq 1$. We consider a continuous operator $M : L_\omega^p(\mathcal{M}, T\mathcal{M}) \to L_\omega^p(\mathcal{M}, T\mathcal{M})$, where $1 \leq p < \infty$. Then,*

$$\forall \phi \in Diff(\mathcal{M}) : \ ML_\phi = L_\phi M$$

*is equivalent to the existence of a scalar $\lambda \in \mathbb{R}$ such that*

$$\forall f \in L_\omega^p(\mathcal{M}, T\mathcal{M}) : \ M[f](m) = \lambda f(m) \quad a.e.$$

We highlight that our theorems are quite generic in the sense that they apply to the manifolds usually used in applications or theory, $\mathbb{R}^d$ in particular.

**Remark 1.** *The scalar case allows to recover standard operators which are exploited for Deep Neural Networks architectures. However, Theorem 2 indicates that the group of diffeomorphism is too rich to obtain non-trivial non-linear operators.*

**Remark 2.** *The case $p = \infty$ leads to different results. For instance, in the scalar case we may consider the operator $Mf(x) = \sup_y |f(y)|$ which fulfills $L_\phi Mf = ML_\phi f$ but is not pointwise.*

**Remark 3.** *The condition "$\omega(\mathcal{M}) = \infty \implies \rho(0) = 0$" in Theorem 1 is necessary, since in the case $\mathcal{M} = \mathbb{R}$, the operator $Mf(x) \triangleq e^{if(x)}$ is not in $L_\omega^p(\mathcal{M}, \mathbb{R})$.*

**Remark 4.** *The Lipschitz condition in Theorem 1 is crucial, otherwise, $Mf(x) = \rho(f(x))$ might not be an operator of $L_\omega^p(\mathcal{M}, \mathbb{R})$. For instance, if $p = 2$, $\mathcal{M} = [0, 1]$ and $Mf(x) = \sqrt{f(x)}$, we see that in this case, let $f(x) = x$, then $f \in L_\omega^p(\mathcal{M}, \mathbb{R})$ and $Mf \notin L_\omega^p(\mathcal{M}, \mathbb{R})$*

**Remark 5.** *If $M$ is not Lipschitz, we can find an example which is not even continuous. The following example holds in both cases, the scalar case and the vector case. In both cases $f \in L^p(M, \mathbb{R})$, the only thing that changes is the action of $L_\phi$ on $f$. $\mathcal{M} = \mathbb{R}$, let for all $f \in L^p(M, \mathbb{R})$:*

$$Mf(x) = 1_{\{z, \lim_{y \to z} f(y) = f(z)\}}(x) f(x).$$

*It is a measurable function. Let us show that this $M$ is a counterexample to the vector case: for any $\phi \in \text{Diff}(\mathcal{M})$ and $x \in \mathbb{R}$, one has*

$$ML_\phi f(x) = 1_{\{z, \lim_{y \to z} f(\phi(y)) = f(\phi(z))\}}(x) \quad d\phi(x)^{-1} f(\phi(x)) \tag{3}$$

$$= 1_{\{z, \lim_{y \to \phi(z)} f(y) = f(\phi(z))\}}(x) \quad d\phi(x)^{-1} f(\phi(x)) \tag{4}$$

$$= 1_{\{z, \lim_{y \to z} f(y) = f(z)\}}(\phi(x)) \quad d\phi(x)^{-1} f(\phi(x)) \tag{5}$$

$$= L_\phi M f(x). \tag{6}$$

*However, $M$ is not continuous as changing any function to $0$ on $\mathbb{Q}$ does not change its norm but changes the set where the limits exists. More precisely let $c > 0$ be a strictly positive scalar, $M[c] = c$; let $f = c1[x \notin \mathbb{Q}]$, $M[f] = 0$ as $\{z, \exists \lim_{y \to z} f(\phi(y))\} = \emptyset$. However $c = f$ almost everywhere but $M[c] \neq M[f]$ therefore $M$ is not continuous.*

## 3.2 Proof Sketch

We now describe the main ideas for proving the Theorems 1 and 2. The appendix contains complete formal arguments and technical lemmata which we omit here due to lack of space. The two proofs share quite some similarities despite substantially different final results. Three ideas guide our proofs: First, we prove that it is possible to localize $M$ on a certain class of open sets which behaves nicely with the manifold structure, the strongly convex sets which we denote as $\mathcal{O}_1$. This is closely related to the notion of pre-sheaf [15]. Secondly, we characterize $M$ on small open-sets. In the scalar case, we will study the representation of locally constant functions. In the vector case, we will show that locally, the image $M(1_U c)$ of a vector field $c$ is co-linear to $c$ provided that $U$ is small enough. We will also show that those local properties are independent of the position on the manifold $\mathcal{M}$ via a connectedness argument. Thirdly and finally, we combine a compacity and a density argument to extend this characterization to $\mathcal{M}$, which is developed in Sec. 3.3. Throughout the presentation, we will use the following definitions and theorems obtained from other works:

**Definition 1** (Strong convexity, from [18]). *Let $\mathcal{O}_1$ be the collection of open sets which are bounded and strongly convex, i.e. such that any points $p, q$ in such a set can be joined by a geodesic contained in the set. Furthermore let $\dot{\mathcal{O}}_1 = \{A \in \mathcal{O}_1 : \exists B \in \mathcal{O}_1, \bar{A} \subset B \text{ and } \omega(\bar{A} \backslash A) = 0\}$.*

The intuition behind the definition of $\dot{\mathcal{O}}_1$ is that all of its elements are contained in a 'security' open set, which avoids degenerated effects on the manifold. In particular, this allows to control the boundary of a given open set.

**Theorem 3** (theorem adapted from [17, 18]). *(1) $\dot{\mathcal{O}}_1$ is a system of neighborhoods. (2) Any element of $\mathcal{O}_1$ is diffeomorph to $\mathbb{R}^d$. (3) Both $\mathcal{O}_1$ and $\dot{\mathcal{O}}_1$ are stable by intersection.*

**Theorem 4** (Flowbox theorem, as stated in [9]). *Let $f, g \in \mathcal{C}_c^\infty(\mathcal{M}, T\mathcal{M})$. For any $m \in \mathcal{M}$ with $f(m) \neq 0$ and $g(m) \neq 0$, there exists an open set $U \subset \mathcal{M}$ and $\phi \in \text{Diff}(\mathcal{M})$ such that $\phi(m) = m$ and $L_\phi(1_U f) = 1_{\phi(U)} g$.*

We will now present some lemmata that are necessary for the proofs of theorems 1 and 2. As a first step, we argue that one may assume $M(0) = 0$ where $0$ denotes the constant $0$-function. This is because in the appendix we show that $M(0)$ is a constant function $C$, with $C = 0$ if $\omega(\mathcal{M}) = \infty$. Therefore, we may substract $C$ from $\rho$ and $\lambda$, leaving us with having to show the theorems only for $M(0) = 0$.

Next, a key idea of the proof is to exploit the flexibility of the deformation equivariance to *localise* the input, i.e. to show that the image of compactly supported functions is also compactly supported. To do so, the following lemma provides a way of collapsing an open ball into a singleton while maintaining a good control on the support of the diffeomorphism.

**Lemma 2** (Key lemma). *Let $\epsilon > 0$. There exists a sequence of diffeomorphisms $\phi_n : \mathbb{R}^d \to \mathbb{R}^d$, compactly supported in $\mathcal{B}(0, 1 + \epsilon)$ such that:*

$$\phi_n(\mathcal{B}(0, 1)) = \mathcal{B}(0, \frac{1}{n}),$$

*and*

$$\sup_{u \in \mathcal{B}(0,1)} \|d\phi_n(u)\| \leq \frac{1}{n} \,.$$

*Proof.* Set $\phi_n(u) = f_n(\|u\|)u$, where

$$f_n(r) = \begin{cases} \frac{1}{n} & \text{, if } |r| \leq 1 \\ 1 & \text{, if } |r| \geq 1 + \epsilon \,, \end{cases}$$

and $f_n$ is smoothly interpolated for $|r| \in [1, 1 + \epsilon]$ in a way that it remains nondecreasing. It is then clear that $\phi_n$ fulfills the desired properties. $\square$

We will often use that if the support of $\phi \in \text{Diff}(\mathcal{M})$ is such that $\text{supp}(\phi) \cap U = \emptyset$, then for any $f \in L^p_\omega(\mathcal{M}, \mathbb{R})$ one has $1_U f = L_\phi(1_U f)$. This implies the following important lemma, for which a rigorous proof can be found in the appendix:

**Lemma 3.** *Let $U \in \dot{\mathcal{O}}_1$ and $M$ as in Theorem 1 or Theorem 2. Then, for any $f \in E$, where $E = L^p_\omega(\mathcal{M}, \mathbb{R})$ or $E = L^p_\omega(\mathcal{M}, T\mathcal{M})$ respectively, we have:*

$$M[f 1_U] = 1_U M[f] \,.$$

*Furthermore, if $U$ is any closed set, the same conclusion applies.*

Equipped with this result, our proof will characterize the image of functions of the type $c 1_U$ where either $c \in \mathbb{R}$, or $c$ is a vector field which can be straightened (isomorphic to a constant vector), via the following Lemma. In the Vector case:

**Lemma 4** (Image of localized vector field)**.** *For $M$ as in Theorem 2 there is $U \in \dot{\mathcal{O}}_1$, and $\lambda(U)$ such that for any $f \in L^p_\omega(M, TM)$:*

$$M[f 1_U] = 1_U \lambda(U) f \,. \tag{7}$$

Here is the scalar case:

**Lemma 5** (Image of constant functions, scalar case)**.** *Let $M$ as in Theorem 1. For any $U \in \dot{\mathcal{O}}_1$ and $c \in \mathbb{R}$, then: $M(c 1_U) = h(c, U) 1_U$. Furthermore, $c \to h(c, U)$ is Lipschitz for any $U \in \dot{\mathcal{O}}_1$.*

At this stage, we note that both representations are point-wise, and the next steps of the proofs will be identical both for the scalar and vector cases. The extension to $L^p_\omega(\mathcal{M}, \mathbb{R})$ or $L^p_\omega(\mathcal{M}, T\mathcal{M})$ will be done thanks to:

**Lemma 6** (Image of a disjoint union of opensets)**.** *Let $U_1, ..., U_n \in \mathcal{O}_1$ and $M$ as in Theorem 2 or Theorem 1, s.t. $\forall i \neq j, \overline{U_i} \cap \overline{U_j} = \emptyset$. Then for any $f \in L^p_\omega(\mathcal{M}, T\mathcal{M})$:*

$$M[\sum_{i=1}^n 1_{U_i} f] = \sum_{i=1}^n M[1_{U_i} f] \,.$$

This lemma states that we can completely characterize $M$ on disjoint union of simple sets. We will then need an argument similar to Vitali covering Lemma in order to "glue" those open sets together, which shows that simple functions with disjoint support can approximate any elements of $L^p_\omega(\mathcal{M}, \mathbb{R})$ or $L^p_\omega(\mathcal{M}, T\mathcal{M})$ (we only state the lemma for $L^p_\omega(\mathcal{M}, \mathbb{R})$ as our proof on $L^p_\omega(\mathcal{M}, T\mathcal{M})$ does not necessarily need this result):

**Lemma 7** (Local Vitali)**.** *For $f \in \mathcal{C}^\infty_c(\mathcal{M})$ and $m \in \mathcal{M}$, there exists $U \in \dot{\mathcal{O}}_1$ with $m \in U$, such that for any $\epsilon > 0$, there exist subsets $U_1, ..., U_n \in \dot{\mathcal{O}}_1$ with $U_i \subset U$ and numbers $c_1, ..., c_n \in \mathbb{R}$ such that:*

$$\| \sum_n 1_{U_n} c_n - 1_U f \| < \epsilon \,.$$

Note that this type of covering is not possible on any open set without further assumptions on the manifold, such as bounds on its Ricci curvature [22]. Fortunately, we will only need a local version which is true because charts are locally bi-Lipschitz. Both Lemma 6 and Lemma 7 imply that:

**Proposition 1.** *Consider $M$ from either Theorem 1 or 2. Assume that there exists $U \in \dot{\mathcal{O}}_1$ such that $M(c1_V) = h(c, V)1_V$ for any $V \subset U$, with $V \in \dot{\mathcal{O}}_1$, where $c$ is either a vector field in the case $E = L_\omega^p(\mathcal{M}, T\mathcal{M})$ or a constant scalar in the case $E = L_\omega^p(\mathcal{M}, \mathbb{R})$. If we further assume that $c \to h(c, U)$ is L-Lipschitz, then*

$$\forall f \in E, \forall m \in \mathcal{M}, M[1_U f](m) = 1_U h(f(m), U).$$

*Furthermore, it does not depend on $U$, meaning that for any other such $\tilde{U}$, we have:*

$$\forall f \in E, \forall m \in U \cap \tilde{U}, M[1_{\tilde{U}} f](m) = 1_U h(f(m), U).$$

We briefly discuss the intuition behind Theorem 2. It is linked to the idea that the operators $M$ at hand have to commute with local rotations, and this even for locally constant vector fields. We reduce the characterisation of deformation-equivariant vector operators using an invariance to symmetry argument: functions which are invariant to rotations are multiples of a scalar. The intuition is contained in the following lemma, which is commonly used in physics:

**Lemma 8** (Invariance to rotation). *Let $f : \mathbb{R}^d \to \mathbb{R}^d$ such that for any $W \in O_d(\mathbb{R})$ and $x \in \mathbb{R}^d$, one has $f(Wx) = W f(x)$. Then, there is $\lambda : \mathbb{R}^d \to \mathbb{R}$, $f(x) = \lambda(\|x\|)x$.*

*Proof.* We write $f(x) = \lambda(x)x + x^\perp$, with $x^\perp(m) \neq 0$ and $x^\perp \perp x$. Then, we introduce $W \in O_d(\mathbb{R})$ such that $W x^\perp(m) = -x^\perp(m)$ and $Wx(m) = x(m)$. From $f(x) = f(Wx) = W f(x)$ we deduce that $x^\perp = 0$. Next, $\lambda(Wx) = \lambda(x)$ thus $\lambda(x) = \lambda(x')$ for any $\|x\| = \|x'\|$. $\square$

**Distinction between scalar and vector case** The scalar case is simpler to handle than the vector case: there are several more steps for the proof of Theorem 2, one needs to show that the point-wise non-linearity is actually a scalar multiplication. We also highlight that the non-linearity is fully defined by its image on locally constant functions.

Finally, we conclude the proof of the theorem by appealing to a common density argument of the functions smooth with compact support, combing all the lemmata we have just presented in Sec. 3.3.

### 3.3 Proofs conclusions (common to the scalar and vector case)

In this section, we prove that the local properties of $M$ can be extended globally on $\mathcal{M}$. The main idea is to exploit the well-known Poincaré's formula, which states that:

$$1_{\cup_i U_i} = \sum_{k=1}^{n} (-1)^k \sum_{i_1 < \ldots < i_k} 1_{U_{i_1} \cap U_{i_2} \cap \ldots \cap U_{i_k}},$$

and to localize the action of $M$ on each $U_{i_1} \cap U_{i_2} \cap \ldots \cap U_{i_k} \in \dot{\mathcal{O}}_1$ thanks to Lemma 3.

*Proof of Theorem 1 and Theorem 2.* Let $f$ be a smooth and compactly supported function. Further consider $\cup_{i \leq n} U_i$ a finite covering of its support with $U_i \in \dot{\mathcal{O}}_1$. Using an inclusion-exclusion formula together with Lemma 3, we obtain

$$1_{\cup_i U_i} M[f] = \sum_{k=1}^{n} (-1)^k \sum_{i_1 < \ldots < i_k} 1_{U_{i_1} \cap U_{i_2} \cap \ldots \cap U_{i_k}} M[f]$$

$$= \sum_{k=1}^{n} (-1)^k \sum_{i_1 < \ldots < i_k} M[f 1_{U_{i_1} \cap U_{i_2} \cap \ldots \cap U_{i_k}}],$$

where we used that $U_{i_1} \cap U_{i_2} \cap \ldots \cap U_{i_k} \in \dot{\mathcal{O}}_1$. Now, the support of $f$ is closed and included in $\cup_i U_i$. Thus using Lemma 3:

$$M[f] = \sum_{k=1}^{n} (-1)^k \sum_{i_1 < \ldots < i_k} M[f 1_{U_{i_1} \cap U_{i_2} \cap \ldots \cap U_{i_k}}],$$

Note that if $\rho$ is a pointwise operator with $\rho(0) = 0$, then $\rho(1_U f) = 1_U \rho(f)$ and

$$\sum_{k=1}^{n}(-1)^k \sum_{i_1 < ... < i_k} \rho(f 1_{U_{i_1} \cap U_{i_2} \cap ... \cap U_{i_k}}) = \sum_{k=1}^{n}(-1)^k \sum_{i_1 < ... < i_k} 1_{U_{i_1} \cap U_{i_2} \cap ... \cap U_{i_k}} \rho(f) \quad (8)$$

$$= 1_{\cup_i U_i} \rho(f) = \rho(f). \quad (9)$$

Thus, $Mf = \rho(f)$ where $\rho$ is obtained from Lemma 4 or 5 combined with Prop 1. We conclude by density in $L_\omega^p(\mathcal{M}, \mathbb{R})$ or $L_\omega^p(\mathcal{M}, T\mathcal{M})$ respectively. This ends the proof. □

## 4 Remarks and conclusion

In this work, we have fully characterized non-linear operators which commute under the action of smooth deformations. In some sense, it settles the intuitive fact that commutation with the whole diffeomorphism group is too strong a property, leading to a small, nearly trivial family of *non-linear* intrinsic operators. While on their own they have limited interest for geometric deep representation learning, they can 'upgrade' any family of linear operators associated with any group $G \subset \text{Diff}(\mathcal{M})$ into a powerful non-linear class — the so-called GDL Blueprint in [4]. Also, this result is a first step towards characterizing the non-linear operators which commute with Gauge transformations and could give useful insights for specifying novel Gauge invariant architectures. We now state a couple of unsolved questions and future work.

**On the commutativity assumption:** Several examples and approximation results [21][35] exist for operators that commute with Lie groups and discrete groups [19]. In this case, it is possible to define a measure on the group that is invariant by the group action (called the Haar measure), which makes it possible to define convolutions. Roughly, non-linear operators covariant with some actions of those groups can be thought of as an approximation by a Group Convolution Neural Networks. It is important to note that the inputs of the operators described in these articles are functions that take real values; the much more general class of inputs that take values in vector bundles is, to our knowledge, not covered in the literature. To our knowledge, we are the first work to study the design of equivariant Neural Networks that process vector fields defined over a manifold. In this setting even for $\mathcal{M} = \mathbb{R}^d$, it is unclear which type of non-linear operators commute with smaller groups of symmetry such as the Euclidean group. In fact, a generic question holds for manifolds: for a given symmetry group $G$, what is elementary non-linear building block of a Neural Network? This could be, for instance, useful to design Neural Networks which are Gauge invariant. It is an open question for future work which would be relevant many applications in physics [16]. Furthermore, the fact that the characterization of diffeomorphism invariant operators we exhibited in this paper is very restrictive opens the way for the study of other non-locally 'smaller' compact groups; we believe that any results in that direction are completely novel.

**Example of vector operators for $L^\infty$** It is slightly unclear how the vector case $p = \infty$ can be handled in our framework, yet [1] seems to have interesting insights toward this direction.

**Linearization of Diff$(\mathcal{M})$** In this work, we considered an exact commutation between operators and a symmetries: however, it is unclear which operators approximatively commute with a given symmetry group. Such operators would be better to linearize a high-dimensional symmetry group like Diff$(\mathcal{M})$. An important instance of non-linear operators that are non-local and that 'nearly' commute with diffeomorphisms is the Wavelet Scattering representation [23, 7, 28].

## Acknowledgments and Disclosure of Funding

EO was supported by the Project ANR-21-CE23-0030 ADONIS and EMERG-ADONIS from Alliance SU. GSP was also supported by France Relance and Median Technologies; he would like to thank very much NeurIPS Foundation for their financial support (NeurIPS 2022 Scholar Award).

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
