# A  Technical Lemmata

*Proof of Lemma 1.* We simply exihibit the proof for $E = L^2_\omega(\mathcal{M}, T\mathcal{M})$. Indeed, let $f \in L^2_\omega(\mathcal{M}, T\mathcal{M})$, then:

$$\|L_\phi f\|^2 = \int g(L_\phi f, L_\phi f) d\omega \tag{10}$$

$$= \int_{\text{supp}(\phi)} g(L_\phi f, L_\phi f) d\omega + \int_{\mathcal{M} \setminus \text{supp}(\phi)} g(L_\phi f, L_\phi f) d\omega \tag{11}$$

$$= \int_{\phi(\text{supp}(\phi))} g(d\phi^{-1}.f, d\phi^{-1}.f) \det(J\phi^{-1}) d\omega' + \int_{\mathcal{M} \setminus \text{supp}(\phi)} g(f, f) d\omega \tag{12}$$

$$\leq \int_{\text{supp}(\phi)} g(f, f) \|d\phi^{-1}\|^2 \det(J\phi^{-1}) d\omega' + \|f\|^2 \tag{13}$$

$$\leq ( \sup_{\omega \in \text{supp}(\phi)} \|d\phi^{-1}(\omega)\|^{2(d+1)} + 1) \|f\|^2 < \infty \tag{14}$$

$$\tag{15}$$

Thus, $L_\phi$ is bounded. □

## A.1  A remark on the Flowbox theorem

Usually, the Flowbox Theorem (here Theorem 4) is stated for a (often local) diffeomorphism. If $c(m) \neq 0, \tilde{c}(m) \neq 0$, then there exists $U, V$ and $\phi : U \to V$ a diffeomorphism such that $m \in U \cap V$ and $L_\phi(1_U c) = 1_V \tilde{c}$. However, we note that thanks to Theorem 4 of [29], it is possible to find $\tilde{U}$ smaller such that there exists $\tilde{\phi} : \mathcal{M} \to \mathcal{M}$ which is a global diffeomorphism and $\forall m \in \tilde{U}, \tilde{\phi}(m) = \phi(m)$. In this case, $\tilde{\phi}, \tilde{U}$ and $\tilde{V} = \tilde{\phi}(\tilde{U})$ are the candidates of our statement in Theorem 4. As this is quite technical and rather intuitive, we skipped this remark in the main paper.

## A.2  Spatial localization (common to the scalar and vector case)

We now explain how to localize our operator $M$. Equipped with Lemma 2, we can extend our contraction result on $\mathbb{R}^d$ to $\mathcal{M}$ as follow:

**Corollary 1** (Contraction of an openset). *For any $U \in \mathcal{O}_1$ and $W$ openset such that $\bar{U} \subset W \subset \mathcal{M}$, there exists $\phi_n$ supported on $W$ such that for any $f \in L^p_\omega(\mathcal{M}, T\mathcal{M})$:*

$$L_{\phi_n}(1_U f) \to 0 .$$

*Proof.* We prove first the result for $U = \mathcal{B}(0, 1)$ and $\bar{U} \subset W$. In this case, it is possible to find $\epsilon > 0$ such that $\mathcal{B}(0, 1 + \epsilon) \subset W$. Now, taking $\phi_n^{-1}$ as in Lemma 2, we get:

$$\int_{\mathbb{R}^d} \|L_{\phi_n}(1_{\mathcal{B}(0,1)} f)(u)\|^p du = \int_{\mathbb{R}^d} \|1_{\mathcal{B}(0,1)}(\phi_n^{-1}(u)) d\phi_n(u).f(\phi_n^{-1}(u))\|^p du \tag{16}$$

$$= \int_{\mathbb{R}^d} \|1_{\mathcal{B}(0,\frac{1}{n})}(u) d\phi_n(u).f(nu)\|^p du \tag{17}$$

$$= \frac{1}{n^d} \int_{\mathbb{R}^d} 1_{\mathcal{B}(0,1)}(u) \|d\phi_n(\frac{u}{n}).f(u)\|^p du \tag{18}$$

$$\leq \frac{1}{n^{d+1}} \|1_{\mathcal{B}(0,1)} f\|^p \to 0 \tag{19}$$

Next, getting back to the manifold, we know that if $U \in \dot{\mathcal{O}}_1$, there is $V \in \mathcal{O}_1$ such that $\bar{U} \subset V$. We can thus find an openset $\mathcal{B} \subset V$, such that in the chart of $V$, $\mathcal{B}$ is an open ball, and $U \subset \mathcal{B} \subset W$. We can thus apply the technique derived above to get $\phi_n : V \to V$, compactly supported, which contracts $\mathcal{B}$(and thus $U$) to 0 and supported in $W$. Since it is smooth, compactly supported on $W$, we can extend it on $\mathcal{M}$ and we get the result. □

Next, this technique can be used to build a sequence of contraction, which allows to explicitly localize the image of a compactly supported function, as follow:

**Lemma 9** ( Lemma 3 restated for closed sets). *Let $F \subset \mathcal{M}$ a closed set. Then, for any $f \in L^p_\omega(\mathcal{M}, \mathbb{R})$, we have:*

$$M[f1_F] = 1_F M[f]$$

*Proof.* Because $\mathcal{M}$ is a manifold, it is second countable and thus there is a countable collection of opens such that $\mathcal{M} \backslash F = \cup_{i \geq 0} U_i$ with $U_i \in \mathcal{O}_1$. We use Lemma 12 and, we apply the dominated convergence theorem to $f_n = 1_{\cup_{i \leq n} U_i} f$ to conclude. $\qquad\square$

*Proof of Lemma 3.* We note that if $U \in \dot{\mathcal{O}}_1$, then $\omega(\bar{U} \backslash U) = 0$ and we can thus use the Corollary 1 to conclude. $\qquad\square$

### A.3 Action on locally constant functions, for the scalar and vector cases

We now prove the part specific to the vector field setting, i.e., that the action of $M$ is locally a multiplication by a scalar.

*Proof of Lemma 4.* **Step 1:** $M(1_U c)(m) = 1_V \lambda(m, U, c)c$ such that $c(m) \neq 0, \forall m \in U$.

Let $c \in C_c^\infty(\mathcal{M}, T\mathcal{M})$. For $U \in \dot{\mathcal{O}}_1$, $m_0 \in U$, fix a chart $\psi : U \to \mathbb{R}^d$, $\psi(m_0) = 0$ and $c$ is constant in $\psi$ denoted $c^\psi \in \mathbb{R}^d$, which is possible thanks to the Theorem 4. This can also be written as for $m$ in a neighborhood of $m_0$:

$$d\psi(m).c(m) = c^\psi \,.$$

Following the strategy in Lemma 8, there is $W \in \mathcal{O}_d$ such that $Wc^\psi = c^\psi$ and $Wv = -v$ for any vector $v$ orthogonal to $c^\psi$. By compacity, we can find $A$ an open set small enough, with boundary of measure 0, such that $0 \in A$, and $\mathcal{W}A \subset \psi(U)$ for any $\mathcal{W} \in \mathcal{O}_d$. Now, setting $\tilde{\phi} = \psi^{-1} \circ W \circ \psi$, which is well defined on the open $\cup_{\mathcal{W} \in \mathcal{O}_d} \mathcal{W}A$, using Theorem 4 of [29](see remark Sec. A.1 of the appendix), we can can extend $\phi$ globally such that on a local neighborhood, $\forall m \in \tilde{U}, \phi(m) = \tilde{\phi}(m)$. Now, up to taking $A$ even smaller, we can use: $V = \overline{\psi^{-1}(\cup_{n \in \mathbb{Z}} W^n A)} \subset U$, which is closed with a measure 0 boundary(we have a countable union). We get:

$$L_\phi(1_V c)(m_0) = [d\psi^{-1}(m_0) \circ W \circ d\psi(m_0)]c(m_0)1_V \tag{20}$$
$$= 1_V c(m_0) \,. \tag{21}$$

Let us denote $p_{c^\psi}^\perp$ the orthogonal projection (with respect to the Euclidean scalar product) on the orthogonal plane to $c^\psi$.

As $V \subset U$, $V$ is closed and $U \in \dot{\mathcal{O}}_1$ from Lemma 9, we know that:

$$M(c)(m_0) = M(1_U c)(m_0) = M(1_V c)(m_0) = \lambda(m_0, c, U)d\psi^{-1}(0)c^\psi + d\psi^{-1}(0)p_{c^\psi}^\perp M(1_V c)(m_0)$$

Yet, on the other hand:

$$L_\phi M(1_V c)(m_0) = \lambda(m_0, c, U)d\psi^{-1}(0)c_\psi - d\psi^{-1}(0)p_{c^\psi}^\perp M(1_V c)(m_0) \tag{22}$$

As this is true for any $m_0$, we thus proved that:

$$M(1_U c)(m) = 1_U \lambda(m, U, c)c$$

**Step 2:** In fact, $\lambda(m, c, U) = \lambda(m, U)$ if $c$ does not cancel on $U$ and $m \in U$.

Let $c, \tilde{c}$ be two vector fields as above and defined on $U$ both not equal to 0, and $m \in U$. Using the Theorem 4 combined with the remark of Sec. A.1 of the appendix, there exists $\phi : \mathcal{M} \to \mathcal{M}$ a diffeomorphism and $\tilde{V}, V \subset U$ and $m \in \tilde{V} \cap V$, such that $L_\phi(1_V c(m)) = 1_{\tilde{V}} \tilde{c}(m)$ and $\phi(m) = m$ Now, we could take a smaller closed set $V \subset U$ with measure 0 boundary, so that $M[1_V c](m) = M[1_U c](m) = M[c](m)$, which would lead to, following a similar argument to above:

$$\lambda(m, \tilde{c}, U)\tilde{c}(m) = M[1_{\tilde{V}} \tilde{c}(m)] = L_\phi M[1_V c](m) = L_\phi(\lambda(., c, U)c)(m) = \lambda(m, c, U)\tilde{c}(m)$$

and then locally $\lambda$ is independent of the choice of a vector field, which implies the desired property.

**Step 3:** In fact, $\lambda(m, U) = \lambda(U)$. Indeed, let $m, m_0 \in V$ and $\phi \in \text{Diff}(\mathcal{M})$ such that $\phi(m) = m_0$ (as $V$ is connex, by using Lemma 11). Now, along the same line as above:

$$\lambda(m, U) = \lambda(m_0, U)$$

The previous results hold when the vector field can be locally straightened, however the vector fields that take value $0$ on some points of $U$ can not be straightened. We will now show that vector fields that can be straightened on $U \in \dot{\mathcal{O}}_1$ are dense dense in $C^\infty(U, TU)$ for the $L^p_\omega$ norm. Let $f \in C^\infty(U, TU)$, let $A = \{x \in U | f(x) = 0\}$, and $A^\epsilon = \{x \in U | \|f(x)\| \leq \epsilon\}$ for $\epsilon > 0$. By Urysohn's lemma there is $\chi^\epsilon : U \to \mathbb{R}$ be such that $\chi|_{A^\epsilon} = 1$ and $\chi|_{U \setminus A^{2\epsilon}} = 0$. Let,

$$f^\epsilon = f + 2\epsilon \chi^\epsilon$$

For any $x \in U$,

$$\|f^\epsilon(x)\| \geq |\|f(x)\| - 2\epsilon \chi^\epsilon(x)| \tag{23}$$

and by construction $|\|f(x)\| - 2\epsilon \chi^\epsilon(x)| > 0$.

Therefore,

$$M[f^\epsilon 1_U] = \lambda(U) f^\epsilon \tag{24}$$

Furthermore for all $0 < \epsilon \leq 1$, $\|f^\epsilon\|$ is bounded by $\|f\| + 2$ that is integrable, so by dominated convergence theorem, $f^\epsilon \xrightarrow[\epsilon \to 0]{L^p_\omega} f$. So, $M[f 1_U] = \lambda(U) f$.

To end the proof, one remarks that $C^\infty_c(\mathcal{M}, T\mathcal{M})$ is dense in $L^p_\omega(\mathcal{M}, T\mathcal{M})$.

$\square$

The next Lemma shows that, in the scalar case, we can consider $\tilde{M} f \triangleq M f - M(0)$ for $f \in L^p_\omega(\mathcal{M}, \mathbb{R})$ without losing in generality.

**Lemma 10.** *Under the assumptions of Theorem 1, $M(0)$ is constant, and if $\omega(\mathcal{M}) = \infty$, then $M(0) = 0$.*

*Proof.* Following the Theorem 1 of [26], for any $m, m_0 \in \mathcal{M}$, we can find $\phi$ global diffeomorphism such that $\phi(m) = m_0$. We note that $L_\phi(0) = 0$ and thus for any $m \in \mathcal{M}$:

$$M(0)(m) = M[L_\phi(0)] = L_\phi M(0)(m) = M(0)(m_0)$$

Thus, $M(0)$ is constant, and if $\omega(\mathcal{M}) = \infty$, it is necessary that $M(0) = 0$. $\square$

The corresponding Lemma in the scalar case is substantially simpler, as strongly convex sets are connex:

*Proof of Lemma 5.* Fix $m_0 \in V$, and let $m \in V$, using Lemma 11(because $V \in \dot{\mathcal{O}}_1$ is connex, we can apply a connexity argument or the transitivity argument of Theorem 1 of [26] for compactly supported diffeomorphisms), we can find $\phi$ supported in $V$ such that $\phi(m_0) = m$. Thus, $L_\phi f = f$ and $M f(m_0) = M L_\phi f(m_0) = L_\phi M f(m_0) = M f(m)$. Thus, $M(c 1_V) = h(c, V) 1_V$. The Lipschitz aspect is inherited from the fact that $M$ is Lipschitz. $\square$

### A.4 Extrapolation to any good open sets (common to the scalar and vector case)

In this section, we use the fact that we want to prove that both scalar and vector operators correspond to point-wise non-linearity, which are locally Lipschitz due to the regularity assumptions that we used.

*Proof of Proposition 1.* **Step 1:Fix** $c$**, for any** $m \in U$ **such that** $V \subset U$**, then** $h(c, U)(m) = h(c, V)(m)$

Indeed, we note that for $m \in U$, where we used Lemma 3:

$$M(1_V f)(m) = 1_V(m)M(f)(m) = 1_U(m)M(f)(m) = M(1_U f)(m)$$

Thus, $h(c, V)|_V = h(c, U)|_V$ for any $V \subset U$.

**Step 2: extension by density, for any** $f$**,** $M(f 1_U) = 1_U h(f, U)$ **for any** $f \in L_\omega^p(\mathcal{M}, \mathbb{R})$. Using Lemma 7, consider $f \in \mathcal{C}_c^\infty(E)$, $f_n = \sum_n 1_{U_n} c_n$, where $c_n$ is either a constant scalar, either a vector field, with disjoint support such that $\|1_U f - 1_U f_n\| < \epsilon$.

We know that, from Lemma 6 that:

$$M(1_U f_n) = M[\sum_n 1_{U_n} c_n] = \sum_n 1_{U_n} M[1_{U_n} c_n] = \sum_n 1_{U_n} h(c_n, U)$$

Next, we note that:

$$\|M 1_U f - 1_U h(f, U)\| \leq \|1_U(M f_n - M f)\| + \|1_U M f_n - 1_U h(f_n, U)\| \tag{25}$$
$$+ \|1_U(h(f_n, U) - h(f, U))\| \tag{26}$$
$$\leq 2L\|1_U(f_n - f)\| \tag{27}$$

and from this, given that $h(., U)$ is $L$-Lipschitz, we conclude by density of $\mathcal{C}_c^\infty(\mathcal{M})$ in $L_\omega^p(\mathcal{M}, \mathbb{R})$.

**Step 3: Independence from** $U$

Step 1 allows for the following definition of a global $h$ from local $h_U$: let $m \in \mathcal{M}$, pose,

$$\forall U \in \dot{\mathcal{O}}_1 \quad h(f(m)) := h(f(m), U) \tag{28}$$

In the scalar case and in the vector case, one can build a scalar function and vector function such that, $f(m) = \mu \in \mathbb{R}$ or $f(m) = c \in T_x M$ (as shown in Step 3 of proof of 4). Therefore in the scalar case $h$ is a function from $\mathbb{R}$ to $\mathbb{R}$ and in the vector case for any $x \in \mathcal{M}$ and $v \in T_x \mathcal{M}$, $h(x) = \lambda x$.

$\square$

We only prove the Vitali version for $L_\omega^p(\mathcal{M}, \mathbb{R})$, as the proof for $L_\omega^p(\mathcal{M}, T\mathcal{M})$ would be identical, replacing solely the scalar by constant vector fields in their local parametrization.

*Proof of Lemma 7.* We consider $U$ small enough such that $U \in \dot{\mathcal{O}}_1$, $m \in U$ and $\exp_m : \mathcal{B} \to U$ is locally a diffeomorphism from $\mathcal{B} \subset T\mathcal{M}_m$, and let $U_i = \exp_m(\mathcal{B}_i)$ with $\mathcal{B}(x_i, r_i) \subset \mathcal{B}$, which is strongly convex and thus $U_i \in \dot{\mathcal{O}}_1$. We remind that $\exp_m$ is bi-Lipschitz on the bounded set $U$. In this case, there is $C_1, C_2 > 0$ such that for any $x_i, r_i$ with $\mathcal{B}(x_i, r_i) \subset \mathcal{B}$, we have $r_i^d \leq \lambda(\mathcal{B}(x_i, r_i)) \leq C_1 \omega(U_i) \leq C_2 \lambda(\mathcal{B}(x_i, r_i)) \leq C_d r^d$. By Vitali's lemma, we have for any $\epsilon > 0$ and $r > 0$, the existence of some $x_i, r_i < r$:

$$\|1_\mathcal{B} - \sum_{i=1}^n 1_{\mathcal{B}(x_i, r_i)}\|^p \leq \epsilon^p$$

For $f$ smooth, let:

$$\|f(x)1_U - \sum_{i=1}^n f(x_i)1_{U_i}\|^p \leq \|\sum_{i=1}^n (f(x) - f(x_i))1_{U_i}\|^p + \|1_{U \setminus (\cup_i U_i)} f(x)\|^p \tag{29}$$

Now, as $\exp_m$ is bi-Lipschitz, we get a $r$ small enough such that $|f(x) - f(x_i)| < \epsilon$. Next, because the sets are disjoint:

$$\|\sum_{i=1}^{n}(f(x) - f(x_i))1_{U_i}\|^p = \sum_{i=1}^{n}\int_{U_i}|f(x) - f(x_i)|^p \tag{30}$$

$$\leq \sum_{i=1}^{n}\omega(U_i)\epsilon^p \tag{31}$$

$$\leq \epsilon^p \omega(U)). \tag{32}$$

Now, using $|f(x)| \leq \|f\|_\infty$, we get:

$$\|1_{U\setminus(\cup_i U_i)}f(x)\|^p \leq \|f\|_\infty \epsilon^p$$

And:

$$\|f - \sum_{i=1}^{n}f(x_i)1_{U_i}\| < (1 + \omega(U))^{1/p}\epsilon.$$

$\square$

The following Lemma allows to build diffeomorphism with compact support - we give this proof for the sake of completeness, at it is proved in [26].

**Lemma 11.** *Fix $\rho > 0$, and $x_0, x_1 \in \mathcal{B}(0, \rho)$, there exists $\phi$ diffeomorphism, such that $\phi(x_0) = x_1$ and $supp(\phi) \subset \mathcal{B}(0, \rho)$.*

*Proof.* Consider $f$, smooth, supported in $[-1, 1]$ and such that $f(0) = 1$. We will use a connexity argument: let us fix $x_0 \in \mathcal{B}(0, \rho)$. Let's consider $\Gamma = \{x \in \mathcal{B}(0, \rho) : \exists \phi \text{ diffeomorphism } \phi(x) = x_0, supp(\phi) \subset \mathcal{B}(0, \rho)\}$. Let $x_1 \in \Gamma$, then there is $\eta < \frac{1}{2}$, $\mathcal{B}(x_1, \eta) \subset \mathcal{B}(0, \rho)$. For $x_2$ such that $\|x_1 - x_2\| \leq \frac{\eta}{4 \sup |f'|}$, we introduce:

$$\tau(x) = (x_2 - x_1)f(\frac{\|x - x_1\|^2}{\eta^2}).$$

We have that $supp(\mathbf{I} - \tau) \subset \mathcal{B}(x_1, \eta)$, and:

$$\frac{\partial \tau}{\partial x}(x) = 2\frac{(x_2 - x_1)\langle x - x_1, x_1\rangle}{\eta^2}f'(\frac{\|x - x_1\|^2}{\eta^2})$$

leading to:

$$\|\frac{\partial \tau}{\partial x}(x)\| < \frac{1}{2}$$

This implies that the spectrum of $\partial\tau$ is in $[0, 1[$ and thus, $\mathbf{I} - \partial\tau$ is invertible. Now, by assumption, we know there is $\phi$ such that $\phi(x_1) = x_0$, compactly supported in $\Omega$. Introducing $\phi' = \phi \circ (\mathbf{I} - \tau)$, then $\phi'$ is a diffeomorphism, compactly supported in $\Omega$ and $\phi'(x_2) = \phi(x_1) = x$, thus $x_2 \in \Gamma$. This shows $\Gamma$ is open. But also $\Gamma$ is closed (otherwiwe, we can make a path ...). Thus, by connexity $\Gamma = \Omega$. $\square$

The next Lemma is crucial in our proof, and allows to characterize union of well behaving opensets:

**Lemma 12.** *Let $n \geq 0$, $\{U_i\}_{i\leq n} \subset \dot{\mathcal{O}}_1$ and $F$ a closed set such that $\bar{U}_i \cap F = \emptyset, \forall i$. Then for any $f \in L_\omega^p(\mathcal{M}, T\mathcal{M})$:*

$$1_F M[(1_F + 1_{\cup_{i\leq n}U_i})f] = 1_F M[1_F f]$$

*Proof.* We work by induction on $n$. For $n = 0$, the result is true. Then, let's write $U_{n+1}^\epsilon = \{x, d(U_{n+1}, x) < \epsilon\}$. It's an openset which contains $\bar{U}_{n+1}$, and by assumption we can pick $\epsilon$ small enough such that $U_{n+1}^\epsilon \cap F = \emptyset$. Next, let's apply Corollary 1 to $U_{n+1}$ and $W = U_{n+1}^\epsilon$. Then:

$$1_F M[(1_F + 1_{(\cup_{i\leq n}U_i\setminus U_{n+1}^\epsilon)\cup U_{n+1}})f] = L_{\phi_n}1_F M[(1_F + 1_{(\cup_{i\leq n}U_i\setminus U_{n+1}^\epsilon)\cup U_{n+1}})f] \tag{33}$$

$$= 1_F M[L_{\phi_n}(1_F f + 1_{(\cup_{i\leq n}U_i\setminus U_{n+1}^\epsilon)\cup U_{n+1}}f)] \tag{34}$$

$$\rightarrow 1_F M[1_F f + 1_{(\cup_{i\leq n}U_i\setminus U_{n+1}^\epsilon)}f] \tag{35}$$

Now, we remark that:

$$1_F M[1_F f + 1_{(\cup_{i \leq n} U_i \setminus U_{n+1}^\epsilon)} f] = 1_F M[1_F f + 1_{\cup_{i \leq n} U_i}(1_{\mathcal{M} \setminus U_{n+1}^\epsilon} f)] \tag{36}$$

And we apply the induction hypothesis to $(1_{\mathcal{M} \setminus U_{n+1}^\epsilon} f)$. $\qquad\square$

The next Lemma is crucial in our proof, and allows to characterize disjoint union of well behaving opensets:

*Proof of lemma 6.* We note that $\cup_{i=1}^n \overline{U_i} = \overline{\cup_{i=1}^n U_i}$. Thus, using Lemma 9, given this union is closed and disjoint and as for any closed set $F$,

$$M[f1_F]1_{F^c} = M[0]1_{F^c} = 0 \tag{37}$$

the following linearity property holds,

$$M[\sum_{i=1}^n 1_{\bar{U}_i} f] = \sum_{i=1}^n 1_{\bar{U}_i} M[f] = \sum_{i=1}^n M[1_{\bar{U}_i} f]$$

Now, we conclude as the boundaries have measure 0. $\qquad\square$