# OpenReview forum: "On Non-Linear operators for Geometric Deep Learning"
_NeurIPS.cc/2022/Conference — NeurIPS 2022 Accept_

### Official Review · Reviewer_TxKA · 2022-07-09

**Rating:** 6
**Confidence:** 2
**Soundness:** 3 good
**Presentation:** 2 fair
**Contribution:** 3 good

**Summary:**

This paper contributes two main proofs that support the practice of using non linear activation functions in conjunction with linear operations in geometric deep learning. They also prove that Diff(M) is too rich and that there are no universal class of non-linear operators to motivate the design of Neural Networks over symmetries of M.

**Questions:**

One thing I suggest to the authors to help with the clarity is to define more of their notation. Much of the notation is not defined in the main paper despite having an entire subsection dedicated to it. This could be improved as there is plenty of space in the paper.



**Limitations:**

The authors discussed some of the unsolved questions that remain.

**Strengths And Weaknesses:**

Strengths: The paper addresses important issues of symmetries in machine learning. It also proves that no universal class of nonlinear operators exist that can handle all possible symmetries in a manifold. This is important to know when designing neural networks. Thus the significance, quality, and originality of the paper are good.

Weakness: The paper is very dense mathematically and difficult to read for someone who is not embedded in the literature on this topic.

---

> ### Author Response · Authors · 2022-08-02
> **Response to Reviewer TxKA**
>
>
> We would like to thank the reviewer for his remarks and positive review! We thought to have added all the definitions to notations once they appeared in the text, yet it seems we might have missed some; we added:
>
> – lines 73 to 76: definition of $L^p(\mathcal{M}, E)$
>
> – line 111: definition of $T_u\mathcal{M}$
>
> – lines 134 to 139: definitions around the tangent bundle, metrics, exterior algebra, sections of a bundle
>
> – lines 140 to 143: general definitions and notations for topological spaces
>
> – lines 154 to 160: definition of the product, $f1_U$, of $f$ and the indicator function on $U$, explanation of the notation $c1_U$ for constant fields over a set $U$.
>
> – line 162: remark on the fact that the reference measure $L^p_{\omega}(\mathcal{M}, E)$ can be sometimes dropped $L^p(\mathcal{M},E)$.
>
>
> If the reviewer can think of one element that he feels to be unclear, we will be extremely happy to add this definition.
>
> We thank the reviewer for this work again.

---

### Official Review · Reviewer_3uBu · 2022-07-11

**Rating:** 5
**Confidence:** 2
**Soundness:** 3 good
**Presentation:** 3 good
**Contribution:** 2 fair

**Summary:**

This paper studies operators mapping vector and scalar fields defined over a manifold. The author demonstrates that the non-linear operators that act on vector fields and commute with the group of diffeomorphisms are scalar multiplications, which implies that $Diff(\mathcal{M})$ is too rich to obtain non-trivial operators. In the case of vector fields $L^{p}_{w}(\mathcal{M}, T, \mathcal{M})$, the author demonstrates that these operators are the scalar multiplication.

**Questions:**

Since I am not an expert in geometric deep learning, I have no more questions about this paper.

**Limitations:**

The limitations are not explicitly discussed in this paper.

**Strengths And Weaknesses:**

As for the strengths of the manuscript, it is well-written and well-motivated. The statements in this paper are well-supported by proof. As for the weaknesses, I prefer to some experiments to verify the proposed the theorems, or provide some examples to show the usage of the proposed methods in the paper.

---

> ### Author Response · Authors · 2022-08-02
> **Response to Reviewer 3uBu**
>
> We thank the reviewer very much for those comments and this view of the paper, which the reviewer believes to be well motivated, written, and well supported.
>
> Thank you for suggesting further examples and experiments. Since our result is essentially an existence result, it is not immediately clear how to illustrate it experimentally. On the other hand, we agree that adding concrete examples of operators used in practice and discussing how they relate to our result will be helpful. In this direction, let us remark that the study of equivariant operators that take as input vector fields is motivated by using neural networks in physics, particularly for dynamical systems such as fluid dynamics [1]. For example, one subject of interest in hydrodynamics is how a vector field of velocities evolves; the time evolution of such field is described by partial differential equations (PDE), the Navier-Stokes equations, in which neural networks found recent applications, and it is more generally the case of other PDE [2]. We added lines 57 to 62 to discuss these examples.
>
> We emphasize that we are, to our knowledge, the first work to study operators that are designed for equivariance on vector fields and not solely signals (taking real values) defined over a manifold. We now discuss the use case of our paper again.
>
> First, some examples and use-cases were already introduced in the introduction of the paper: our work indicates that as the group of diffeomorphisms is not locally compact, it is impossible to find a universal non-linearity that could be used in any neural networks for manifolds. This contrasts with the case of the affine or euclidean group of transformation. A practitioner must, for now, rely on a complex routine of trial and error to find an appropriate non-linearity for designing a neural network if no additional regularity, assumption, or symmetry group is known about this manifold structure. Our work proposes a class of non-linear operators that satisfy some basic assumptions and can be automatically considered by a practitioner.
>
> We believe this to be difficult for an experiment, as our result is mainly an existence result. Furthermore, we note that working directly with a symmetry group on a discretized grid is difficult. Indeed, naively discretizing a symmetry group reduces the action of the permutation group on a vector space that is not rich enough to capture the richness of the group of diffeomorphisms. This makes the numerical experiments challenging. However, we, too, would be very interested in doing some numerical experiments in this setting, and if the reviewer has some method to propose, we would be glad to try it.
>
>
>
> We respectfully disagree that the limitations of the paper are not discussed in the paper: in Sec. 4, we discuss several limitations (extension to $L^p$, $p=\infty$, the notion of Gauge transform that should be addressed in future works). Also, Remarks 1-5 of Sec 3.1 discuss our theorems' hypothesis, show several counter examples and challenge our hypothesis.
>
> We thank the reviewer for this work again.
>
>
> [1]S.L. Brunton, B.R. Noack  and P. Koumoutsakos, Machine Learning for Fluid Mechanics, Annual Review of Fluid Mechanics, 2020.
> [2]M. Raissi, P. Perdikaris and G.E. Karniadakis, Physics-informed neural networks: A deep learning framework for solving forward and inverse problems involving nonlinear partial differential equations, Journal of Computational Physics, 2019.

---

### Official Review · Reviewer_YVG6 · 2022-07-15

**Rating:** 7
**Confidence:** 2
**Soundness:** 3 good
**Presentation:** 3 good
**Contribution:** 3 good

**Summary:**

This paper theoretically studies the operators on a manifold $\mathcal{M}$ that commute with the group of diffeomorphisms $\text{Diff}(\mathcal{M})$. Specifically, the author proves that in the case of a scalar fields, only point-wise non-linearities are the corresponding operators; and in the case of vector fields, those operators corresponds to the scalar multiplication. These result indicates that there is no universal class of non-linear operators (e.g., neural networks) that commute with any group of symmetries of $\mathcal{M}$

**Questions:**

All the suggestions and questions are presented in the strength and weaknesses section.

**Limitations:**

Yes

**Strengths And Weaknesses:**

Strengths:
- This paper, especially the theory and its corresponding proofs, is well-written and easy to follow.
- The proved theory appears to be novel and solid, especially the result in the context of neural networks. I think this is a very nice contribution to the geometric deep learning theory.

Weaknesses and comments:
- Overall, I don't have any criticism on the theories, my only question is that since one of the result is that there is no universal class of non-linear operators that commute with 'any' group of symmetries, I wonder what can we say about the operators if we reduce the symmetries to some specific symmetry group? In this case, would that be possible to find some particular non-linear operators that commute with such symmetries? I think it would be great if there is a discussion on this.

Minor comments:
- The operator $M$ (not the manifold $\mathcal{M}$) is mentioned in line 16 but it is defined later in line 64.

---

> ### Author Response · Authors · 2022-08-02
> **Response to Reviewer YVG6**
>
> We would first like to thank the reviewer for this positive review, these remarks, comments, and interest in this work. First, we have added a sentence on line 15 to define operator M, as noted.
>
> Knowing if there are smaller groups for which one can find non-linear operators that commute with the elements of this group is very natural, especially after this no-go theorem: here, we propose to study the 'biggest group' one can consider (the group of diffeomorphisms).  Some results on non-linear operators that commute with locally compact groups exist. However, we are unaware of similar results for non-locally compact groups, of which the group of diffeomorphisms is an example. In this respect, the results of our paper are a first step toward the study of nonlinear operators that are covariant with ‘large’ non-compact groups of symmetry and points out the necessity to characterize operators that are covariant to ‘smaller’ ‘large’ groups.
>
> To complete the previous remarks, we’d like first to highlight that in the non-linear case, several results [1,2] exist for the Lie group setting and discrete groups [3]: roughly, non-linear operators covariant with some action of those groups can be thought of as an approximation by a Group Convolution Neural Networks.  However, our current work aims to go beyond the Lie group setting, which we believe to be easier than the generic case because Lie groups are embedded with a low-dimensional Euclidean structure. More generically, we note that once a group is locally compact (e.g., Lie groups), it is possible to define a Haar measure. From this stage, it is possible to describe the class of linear operators which commute with this group. Note, however, that we do not know about the classification of the nonlinear operators that commute with locally compact groups when the input data is not a function that takes values in $\mathbb{R}$ but a section of a vector bundle. Furthermore, we do not know about results in the direction of non-compact groups (e.g., the group of diffeomorphisms $\text{Diff}(\mathcal{M})$). We leave this exploration for future work.
>
> Another direction is to change the functional space on which the group acts, and indeed, here we consider $L^p$ spaces; however, we suspect the characterization of non-linear representation that commutes with the group of diffeomorphisms to be much richer and much more difficult when considering $C^\infty$ or even solely continuous functions.
>
> We have added a paragraph from lines 293 to 302 in the Remarks and conclusion section to discuss these important points
>
> Ref:
> [1] Kumagai, W. and Sannai, A., Universal approximation theorem for equivariant maps by group cnns. arXiv preprint arXiv:2012.13882,  2020.
> [2] Yarotsky, D., Universal Approximations of Invariant Maps by Neural Networks, Constr Approx, 2022.
> [3] N. Keriven and G. Peyré, “Universal invariant and equivariant graph neural networks”, Advances in Neural Information Processing Systems, 2019.

---

### Author Response · Authors · 2022-08-02
**General response to all reviewers**

We thank all the reviewers for their work and comments on our mathematical sound contribution. We’d like to highlight that despite being quite theoretically grounded, the reviewers have generally acknowledged that we had a good and valuable contribution. During the discussion, we will be extremely happy to discuss any mathematical elements that might have been unclear, as we understand that certain elements might be technical. We thank all the reviewers again for their comments which we believe to be useful to improve the overall quality of this paper. We now answer each reviewer individually.

---

### Meta-Review · Area_Chair_YMFq · 2022-08-23

**Recommendation:** Accept
**Confidence:** Less certain

**Metareview:**

Overall: studies operators mapping vector and scalar fields defined over a manifold M, and which commute with its group of diffeomorphisms Diff(M).

Reviews: The paper received three reviews, all of them on the positive side: Accept (less confident), Weak Accept (less confident), borderline accept (less confident). It seems that all reviewers are happy with the paper + the proposed changes by the authors during the rebuttal.. The reviewers found the paper is clear and has a clean presentation. The findings are interesting, and connect with the ML community. The authors have provided satisfactory answers to reviewers' comments, answering most of them successfully.


Confidence of reviews: Overall, the reviewers are fairly confident. We will put more weight to the reviews that got engaged in the rebuttal discussion period.

**Award:**

No

---

### Decision · Program_Chairs · 2022-09-14

Accept